# Patient-Derived Xenograft vs. Organoids: A Preliminary Analysis of Cancer Research Output, Funding and Human Health Impact in 2014–2019

**DOI:** 10.3390/ani10101923

**Published:** 2020-10-20

**Authors:** Lindsay J. Marshall, Marcia Triunfol, Troy Seidle

**Affiliations:** 1Humane Society International and the Humane Society of the United States, Washington, DC 20037, USA; 2Humane Society International, Washington, DC, 20037, USA; mtriunfol@hsi.org (M.T.); tseidle@hsi.org (T.S.)

**Keywords:** patient-derived organoids, patient-derived xenografts, breast cancer, colorectal cancer, lung cancer, research outputs

## Abstract

**Simple Summary:**

Given ongoing issues with public opinion against the use of animals in biomedical research, scant research resources, high drug costs, low translational success rates, and the length of time that it currently takes for new drugs to reach the market, we undertook a preliminary analysis of cancer research involving animals, compared to studies using non-animal, human-relevant approaches, carried out between 2014–2019. We collated research studies in the United States and across the European Union, which had used animal models or non-animal, patient-derived cell-based models to study breast, lung and colorectal cancers, and looked for evidence of impacts of this research on human health. We noticed that the number of publications focused on human organoid models shows a growing trend over time. We also saw that there is evidence of increasing funding support for the human cell-based methods, although these still lag behind the animal-based research. Overall, we found that more could be done to promote the use of human-relevant methods and we call for the major funding organizations to prioritize the use of non-animal methods.

**Abstract:**

Cancer remains a major threat to mortality and morbidity globally, despite intense research and generous funding. Patient-derived xenograft (PDX) models—where tumor biopsies are injected into an animal—were developed to improve the predictive capacity of preclinical animal models. However, recent observations have called into question the clinical relevance, and therefore the translational accuracy, of these. Patient-derived organoids (PDO) use patient tumor samples to create in vitro models that maintain aspects of tumor structure and heterogeneity. We undertook a preliminary analysis of the number of breast, colorectal, and lung cancer research studies using PDX or PDO published worldwide between 2014–2019. We looked for evidence of impacts of this research on human health. The number of publications that focused on PDO is gradually increasing over time, but is still very low compared to publications using PDX models. Support for new research projects using PDO is gradually increasing, a promising indicator of a shift towards more human-relevant approaches to understanding human disease. Overall, increases in total funding for these three major cancer types does not appear to be translating to any consequential increase in outputs, defined for this purpose as publications associated with clinical trials. With increasing public discomfort in research using animals and demands for ‘alternative’ methods, it is timely to consider how to implement non-animal methods more effectively

## 1. Introduction

Despite recognition of the condition for many centuries [1], intense research efforts and generous funding, cancer remains a major threat to mortality and morbidity globally [2]. Across the world, cancer was responsible for more than 9.5 million deaths in 2018, and more than 18 million new cases were recorded in the same year [3]. According to the Globocan database (details in Table 1), the five types of cancer with the highest mortality (deaths recorded in 2018) are as follows: lung (1,761,007), colorectal (880,972), stomach (782,685), liver (781,631) and breast (626,679). Whilst there are regional differences in incidence and mortality, Europe, the UK and the US follow this global trend, with lung, breast, and colorectal cancers accounting for the top three cancer-related mortalities.

Patient-derived xenograft (PDX) models—where tumor cell lines or patient biopsies are injected into immunodeficient or genetically modified animals—were originally developed with the rationale of improving the predictive capacity of preclinical animal models, and thereby improving translation rates of oncology treatments. The advantages of this whole animal model were purported to include the ability to mimic a systemic response, the ability to model metastasis and the maintenance/development of an intact tumor microenvironment. However, the use of immunocompromised mice permissive to tumor growth also compromises the immune response hindering effective immunotherapy testing, spontaneous PDX metastases occur very rarely [4] and this tumor microenvironment is based on (immunocompromised) mouse biology and physiology rather than human. Therefore, PDX models fail to accurately represent the human situation, are time-consuming and costly and fail to offer a route for high-throughput screening. Furthermore, PDX models consume large numbers of animals in an era where there is intense, justified pressure to reduce and replace animal use. Thus, there remains a need to develop patient-specific models that maintain aspects of tumor architecture that allow personalized, combination drug screening, and permit timely and cost-effective drug screening.

Several recent observations using the animal-dependent PDX models have called the clinical relevance, and therefore the translational accuracy, of this approach into question. For example, PDX models are recreated from small samples of the overall tumor, and it has been shown that less than half of the full range of mutations are represented in the PDX model, and that additional mutations may appear during culture [5]. Analysis of over 1000 PDX models, derived from 24 different cancers, demonstrated dynamic alterations in tumor genetics over time, such that the “genetic composition of a PDX tumor may differ from that of its matched primary tumor, potentially in therapeutically meaningful ways” [6]. Moreover, there may be clonal selection of the xenograft, with only the less differentiated cells growing in the animals, and there is gradual replacement of the (human) tumor stromal components with mouse fibroblasts, endothelial cells, and inflammatory cells. There also remain the insurmountable species differences between mouse and human that have been shown to impact translational efficacy for many diseases [7] and which can have particular impact in assessing drug pharmacokinetics. When it comes to testing drugs or combinations in order to derive personalized treatment regimes, PDX models may require several months to reach an appropriate size—which does not match the required clinical timeframe [8].

Organoids were first developed in 2009 [9]. These are miniature three-dimensional organ constructs, derived from stem cells and grown in vitro as self-organizing structures that retain features of the native organ. Tumor organoids, or patient-derived organoids (PDO), are ones which use patient tumor samples to create models that maintain aspects of the tumor structure and heterogeneity. PDO have been shown to recapitulate various key features of the original cancer, including the multicellular nature and histo-architecture, together with intercellular communications, and genetic diversity (including intra-tumoral heterogeneity), and therefore offer a more physiologically, pathologically realistic platform against which to screen drugs/combinations than two-dimensional cultures or cancer cell lines [10,11,12]. In addition to this, PDO are amenable to long-term expansion, high-throughput drug screening, and even cryopreservation, offering important advantages over the PDX models, and enabling the creation of human tumor biobanks [13]. Studies comparing PDO and patients indicate their impressive predictive abilities: for each of 55 drugs tested on organoids, a significant impact on the organoid was only seen in those PDO derived from patients who had responded to the drug [14]. This accurate simulation of cancer behavior ex vivo puts PDO at the forefront for decision-making processes prior to clinical trials.

## 2. Materials and Methods

We undertook a preliminary analysis of the number of PDX cancer research studies involving animals that were published worldwide over the period 2014–2019. In our preliminary analysis, we also collated National Institutes of Health (NIH)-funded studies that used PDX models to study the major cancers affecting the developed nations (breast, lung, colorectal), and looked for evidence of impacts of this research (i.e., clinical trials associated with the research). We went on to carry out a similar analysis for PDO-dependent publications of the same cancer types. We used PubMed’s ‘Clinical Trial’ filter (type of study) to see how many of the published papers were associated with clinical trials, and we used the NIH RePORT database to further search for clinical trials that are using organoids or xenografts. We collated the available data on projects funded by the NIH and through the European framework programs to estimate the prevalence of PDX versus PDO in research projects. Note that we used the publicly available abstracts to gauge the use of PDX or PDO in the research, and so we cannot definitively state that a project is or is not using animal or non-animal approaches.

All searches were performed on 8 August 2020 using the updated version of PubMed (Appendix A has a list of all queries and all the results retrieved using each one). Note that we excluded reviews from our searches, as our interest was to retrieve primary research on these topics. This decision may have excluded some primary research that was eventually published together with a review and classified by PubMed as a review.

We first asked how many papers were published between 2014 and 2019 to study each of the three cancers (breast, lung and colorectal cancers) regardless of the technology or approach used in order to find out the general trend in these areas of research. For this, we used the following search string:

(breast cancer[MeSH Terms]) AND ((“2014”[Date-Publication]: “2019”[Date-Publication]))) NOT ((review[Publication Type]OR(systematic review[Publication Type]))

We used the same strategy to search for studies on lung cancer and colorectal cancer.

In order to find publications using PDX, we used the following strategy, for each type of cancer (breast cancer is shown below as an example):

((breast cancer[MeSH Terms]) AND (Xenograft Model Antitumor Assays[MeSH Terms]) AND (“2014”[Date-Publication]: “2019”[Date-Publication])) NOT ((review[Publication Type]) OR (systematic review[Publication Type]))

Note that we used the MeSH Term “Xenograft Model Antitumor Assays”, defined in PubMed as “in vivo methods of screening investigative anticancer drugs, biologic response modifiers or radiotherapies. Human tumor tissue or cells are transplanted into mice or rats followed by tumor treatment regimens. A variety of outcomes are monitored to assess antitumor effectiveness”. This definition aligns with that of PDX in cancer research and thus, in this study, we refer to these animal-based models as PDX.

To find publications on PDO, we used the following strategy, for each type of cancer (breast cancer is shown below as an example):

((breast cancer[MeSH Terms]) AND (organoids[MeSH Terms]) AND (“2014”[Date-Publication]: “2019”[Date-Publication]) AND (humans[MeSH Terms])) NOT ((review[Publication Type]) OR (systematic review[Publication Type])).

The MeSH Term ‘organoid’, defined in PubMed as “an organization of cells into an organ-like structure. Organoids can be generated in culture. They are also found in certain neoplasms” is referred to throughout this study as PDO.

For studies undertaken in the United States, we used the filter option ‘Affiliation’ available in PubMed. It should be noted that not all studies selected using this strategy were actually carried out in the United States. As long as at least one of the study’s authors is affiliated with a scientific institution located in the US, the study will be retrieved in the search. Below is the search string used to retrieve publications in the United States on PDX in breast cancer, as an example:

((breast cancer[MeSH Terms]) AND (Xenograft Model Antitumor Assays[MeSH Terms]) AND (“2014”[Date-Publication]: “2019”[Date-Publication]) AND ((“United States”[Affiliation] OR “US”[Affiliation] OR “USA”[Affiliation] OR “United States of America”[Affiliation]) NOT ((review[Publication Type])OR(systematic review[Publication Type])).

We then queried how many of these were funded by NIH:

((breast cancer[MeSH Terms]) AND (Xenograft Model Antitumor Assays[MeSH Terms]) AND (“2014”[Date-Publication]: “2019”[Date-Publication]) AND ((“United States”[Affiliation] OR “US”[Affiliation] OR “USA”[Affiliation] OR “United States of America”[Affiliation]) AND (“nih funded”[Filter])) NOT ((review[Publication Type]) OR (systematic review[Publication Type])).

Here, we cannot state that studies returned as NIH-funded were only funded by NIH, as other funding sources may have contributed to the research, and we cannot make any statements about which elements of the paper were specifically associated with NIH funding.

We applied the same strategy to retrieve publications using PDO. It should be noted that the number of year by year publications may show some duplicity because the year timeline counts all publication dates for a citation as supplied by the publisher, such as print and electronic publication dates, and these dates may span more than one year, as explained in the PubMed user guide [15].

All information retrieved from PubMed was compiled in a spreadsheet and is available as Appendix A. We only performed simple calculations that did not required any specialized software.

We used the NIH RePORT database (Table 1) to examine research funding for NIH research project grant programs, classified as R01 projects [16], in their first year, that contain the keyword xenograft, or organoid. We filtered results based on active projects in each fiscal year between 2014 and 2019, to retrieve a year by year list of grants.

In order to obtain an indication of European funding, since it is not possible to extract these data from PubMed, we used the CORDIS database (Table 1) and searched for either breast cancer and xenograft, or breast cancer and organoid. We checked every project and discarded those which did not refer to breast cancer, xenografts, or organoids. We then calculated the total funding for projects, focusing on those that were funded through Horizon 2020 or European Research Council. We repeated this search to retrieve projects using either xenografts or organoids for lung cancer and colorectal cancer.

To look at UK and US funding, we accessed the UK National Cancer Research Institute (NCRI) and the US NIH Databook (Table 1), respectively, to retrieve funding data for breast, colorectal and lung cancers. We extracted the funding data for the annual spend for each cancer type and the total spend for all sites to enable calculation of the relative spend per cancer type, per year, between 2014 and 2019.

We used the 2019 report on the use of animals for scientific purposes in the Member States of the European Union [17] to calculate the number of animals used for cancer research for the years 2015, 2016 and 2017. We focused on mice, as these were the majority of PDX models that our previous research had revealed, although we also noted that rats are used for PDX. To estimate animal/mouse use for biomedical research (and omit regulatory purposes for which there is a legal requirement) we totaled animal/mouse numbers returned in the categories of ‘Basic research’ and ‘Translational and Applied Research’. To estimate animal use for cancer research in the European Union, we used the data returned in the categories of ‘Human cancer’ and ‘Oncology’. We appreciate that this may underestimate animal use for cancer or related areas such as tumor development or chemotherapies more broadly, since additional categories may capture these uses.

## 3. Results

We searched for the total number of published studies worldwide that used either PDX or PDO to study three of the cancers with the highest mortality, i.e., breast, lung and colorectal cancers. We then filtered these results to determine the subset of these publications that had some association with a clinical trial, as an initial indicator of research impact on human health.

We first established the total number of studies published for each of the three types of cancer, regardless of the technology or approach used, during the period 2014–2019 to envisage the global trend during this time period. We observed that in all three cases, the number of publications slightly decreases over the period of investigation (Figure 1A). We next examined the proportion of these publications associated with a clinical trial, over the same period. For all three types of cancer, the number of publications associated with a clinical trial also drops over time (Figure 1B).

We calculated the relative proportion of publications for each type of cancer relative to the total number of papers (Figure 2), separating the technologies so that we could understand the extent to which PDX are used, compared to PDO. These data indicate that PDX were the preferred technology for each type of cancer studied here, and therefore there were more clinical trials associated with papers using PDX.

We next investigated the number of published studies using PDX or PDO for each individual year between 2014 and 2019, and also retrieved those publications associated with a clinical trial for the three types of cancer studied here, for each technology (Figure 3).

These data indicate that is no obvious trend in the number of papers using PDX for any of the cancer types (Figure 3A), or in the number of publications associated with clinical trials (Figure 3B), suggesting there was no appreciable increase in the application of the PDX technology to clinical trials for this time period. For PDO, a relatively new technology, we see a different trend (Figure 3C): For all three cancers studied, we see a steep increase in the number of papers published year on year, with all three reaching a maximum number in 2019, returning 28, 14 and 88 publications for breast, lung and colorectal cancers, respectively, during the period. We saw a reversal of this trend for colorectal cancer—we noted fewer papers using PDX for this cancer type, compared to breast or lung cancer, whereas the number of papers published using PDO for colorectal cancer was consistently higher than either breast cancer or lung cancer for each year that we examined. We did not retrieve any papers using PDO and associated with clinical trials for any of the cancers selected here.

We then set out to examine how this global trend compares to any trend observed in the United States, how many studies have received NIH funds, and the proportion of these that are associated with a clinical trial. Table 2 summarizes these data. For research using PDX models, we found some association between the total number of publications and clinical trials—there were more papers for lung cancer, followed by colorectal cancer, then breast cancer, and where we saw more publications for a particular cancer type, there were more clinical trials.

For studies employing PDO, there are far fewer papers than we retrieved for studies using PDX, for all cancer types, and this was true for both total studies and NIH-funded studies. It is perhaps not surprising that there were fewer clinical trials associated with these publications—in fact, we only found one clinical trial that was using PDO, this was for lung cancer.

The number of articles with an affiliation to the United States, in which researchers were using PDX to study any one of the three cancers investigated here, closely mirrors the global trend, and the numbers show that most published studies in this field are performed in the United States or have collaborators located in American institutions. Like the publication trend worldwide, in the US, we also see some small fluctuations in the number of studies published but overall these numbers have not changed much during the period (Figure 3A and Figure 4A). We observed that the proportion of studies associated with clinical trials using PDX in the United States generated a result very similar to that observed for the global trend, in which no trend can be identified, and no growing trend is observed (Figure 3B and Figure 4B). For published studies with an US affiliation, there were higher numbers of papers using PDX for research into breast and colorectal cancers. This trend was also observed worldwide (Figure 3A). For PDO, there were fewer papers retrieved for all cancers, both worldwide and for those with a US affiliation, perhaps reflecting the novelty of this methodology, at least compared to animal models. However, for papers using PDO, there is a growing trend in their use for all cancers over time in the US, except for colorectal cancer where we see a small drop between 2018 and 2019 (Figure 4E). In any case, colorectal cancer is the one type of cancer where we see the highest number of papers published overall (88 papers globally, 25 in the US), indicating a great interest in using PDO to study this type of cancer. (Moreover, it should be noted that in this case, the reality is that the drop in the number of papers between 2018 and 2019 for papers published in the US is due to one paper as a consequence of a duplication in the year by year count, generated by PubMed and as explained in the Materials and Methods section).

### 3.1. Cancer Research Funding Is Maintained over Time

To understand how the overall landscape of cancer research has changed over time, we extracted funding data from the UK National Cancer Research Institute (NCRI) and the US NIH (Figure 5). For both the UK and US, breast cancer research is relatively well funded in comparison to either colorectal cancer or lung cancer (Figure 5A,B), with an average of around GBP 69 million or USD 485 million supporting breast cancer annually for the UK and US, respectively. Figure 5A,B show that, for breast and colorectal cancers, funding levels have not altered greatly over time, with slight increases in funding that are not linked to inflation. Lung cancer has received more support over time, with an increase of 60% since 2014 in the UK and 40% in the US, in the face of overall increases in total cancer funding of around 20% in both the UK and US. In the UK, research funding allocated to breast cancer represents at least 20% of total cancer research funding, with colorectal cancer and lung cancer attracting around 15% of the total budget (Figure 5C), and this apportionment has stayed steady over the period of investigation, even as total funding for all three cancer types has increased over time. The same overall trend is true for the US for each cancer type (Figure 5D), although the funding relative to total cancer spend is lower in the US, with breast cancer at around 8% and lung/colorectal cancer less than 5% each. We must stress that these are preliminary analyses that take a broad overview of the funding for cancer, but the data do indicate that cancer research is relatively well supported and continues to be so over time. For example, the total funding budget for the NIH in 2019 was USD 39.1 billion [18], of which USD 6.25 billion (16%) was invested into cancer research, and USD 701 million specifically to breast cancer research.

### 3.2. Is the Picture Changing with Newly Funded Research?

For NIH research, we were interested in how many applications for novel research projects (first year one of R01 projects) were funded using either mouse PDX models or PDO, to see if there is a trend towards increasing support of new research using the more human-relevant and human predictive technologies. The data in Table 3 show that successful new applications for research into breast cancer using the organoids are very low and have been for the five years in which we searched. For 2019, there were only two projects that were dedicated to the organoids without incorporating animal models, attracting a total of less than USD 2 million. In contrast, support of research projects using xenografts is increasing, from 17 projects and a total funding value of USD 6.9 million in 2014, to 23 projects with a total funding value of almost USD 11 million in 2019. We noted that many grants used multiple approaches, including combinations of in vivo, in vitro and clinical samples and for three grants overall, organoids and xenografts were described in the same application.

### 3.3. European Funding Supports Animal-Based Research

We used the CORDIS database (Table 1) to give us a snapshot of recent EU-funding for breast, colorectal and lung cancer research. This revealed a very similar picture to the US and UK, in that the animal-based methods are supported more than non-animal (organoid) approaches. For breast cancer, our search returned 13 projects that mentioned either PDX models or genetically modified mouse models for breast cancer. These projects all started on or before 2015, with a total funding value of over EUR 18 million and an average award of around EUR 1.5 million. In contrast, projects declaring use PDO without any animal methodologies were fewer and attracted less financial support. We found only five projects that were focused on organoid methods. These projects were awarded a total of EUR 4.7 million, with an average value of EUR 0.9 million per project. Interestingly, the projects that were using organoids, in the absence of any animal models, were all started in or after 2018, whereas we noted that projects returned with the search term organoid prior to this data also employed animal models.

We were surprised to find that there were no Horizon 2020-funded projects using mouse xenografts for lung cancer and only one European Research Council-funded project with a funding commitment of less than EUR 150,000. We did retrieve two projects aiming to use organoids for lung cancer research, which are receiving a total of EUR 2.5 million. For colorectal cancer, the total funding for fifteen projects using xenografts was around EUR 32 million, but interestingly, we noted that six of these projects described their intention to use organoids alongside the animal-based methods, but did not describe this purpose. A further six projects appeared to be dedicated to organoids, with no mention of animal models, and these attracted a total of EUR 6 million. This apparent interest in the use of patient-derived organoids for colorectal cancer research fits with our findings regarding the relatively high number of publications on this cancer type using PDO technology, both globally and for the US (Figure 3C and Figure 4D).

### 3.4. Animal Use for Cancer Research Is Sustained over Time in the European Union

Table 4 shows our calculations for the number of animals, and the numbers of mice, used for research and testing, the numbers used specifically in basic, translational and applied research, and the percentage which are used specifically in cancer research. Use of animals for scientific purposes is not declining, with more than 9.5 million instances of animal use every year for 2015, 2016 and 2017. Animal use for basic and translational and applied research has remained similarly static, accounting for around 70% of the total animal use. Mice bear the burden of biomedical research-calculated as the total use for basic research and translational and applied research—and also bear the brunt of cancer research. The number of mice used in basic and translational research is around 4.5 million, representing around 70% of the total animals used for these purposes. Furthermore, mice make up over 90% of the animals used for cancer research and have done so for 2015, 2016 and 2017, representing around 1 million animals a year.

## 4. Discussion

We found evidence of increased use of human-relevant organoids in research, and for exploring personalized medicine approaches clinically. However, it is of concern that over five times more NIH R01 funding for new research grants appears to be dedicated to PDX than PDO. Given the promise of human-derived organoids, and their increasing success with defining drug combinations and personalized regimes for people with cancer, we suggest that judicious funding calls promoting the use of human-specific methods such as the PDO would offer a greater return on investment, and a pathway to personalized medicine for all.

We also analyzed the published papers on the selected cancer types were associated with a clinical trial using a filter available in PubMed that selects clinical trial as a type of paper. It should be noted that the papers selected using the clinical trials filter are not necessarily studies reporting on results of clinical trials. These studies may be extended analyses of a subgroup previously analyzed in a trial or may be a report on a specific patient who was part of a clinical trial, or may be a study doing additional testing of a drug that was tested in a clinical trial. Thus, these studies should be better considered as studies that are associated with clinical trials, although the extent of such association is not always clear.

When calculating the total amount of funding per year for each of these cancers, we were unable to apportion precise amounts of support to xenografts or organoids, as the databases we used do not define which projects used animals or are animal-free. These data are not always captured in the databases or are not publicly available. The funding data offer a broad snapshot of where these funds are being invested, but we cannot use these to comment on the relative support for non-animal methodologies.

Here, we focused on the three types of cancer with the highest mortality rates for the US and EU [3] and these may be more amenable to organoid models offering predictions for possible treatments or manipulation of disease progression. A recent review in Nature compared PDO with other model systems [11], rating organoids highly for ease of establishment, maintenance, genetic manipulation, genome wide screening and recapitulation of human physiology. However, we note that this review could only describe PDO as ‘partly suitable’ in terms of their ability to recapitulate physiological complexity. It is apparent that organoid models for more complex cancers such as glioblastoma offer a greater challenge. Perhaps targeted funding calls that promote development or use of the non-animal, human relevant methodologies would offer the first step toward better understanding of what obstacles remain for effective cancer interventions, e.g., cellular invasion to disparate organs throughout the body?

We cannot make any assumptions about the proportion of funding that is allocated to animals versus non-animal approaches here. The data do not allow us to see any shifts in support toward the non-animal technologies and away from animal models; indeed, our research indicates that often a variety of methods are employed on any one project and these may include both animal and non-animal techniques.

Our research revealed that the total funding available for the three major cancer types affecting the US and EU has undergone a modest increase year-on-year since 2014, but this does not appear to be translating to any meaningful increases in output, as we saw no appreciable difference in the number of papers published (a metric frequently used to quantify research impact [19]) or the number of papers associated with clinical trials. Whilst our conclusions are necessarily limited, and we cannot accurately calculate the proportion of funding allocated to projects dedicated to the newer, human-relevant technologies as total replacement for animals, we did see that, for NIH projects funded in 2019 alone, R01 research projects using mouse xenograft models for breast cancer research attracted over five times greater support than projects using organoids to explore breast cancer (Table 3). It is concerning that research projects using animal models continue to attract substantially more funding than human-based methods, and that this sustained reliance on animal models is associated with a less-than-optimal return on investment [20]. We suggest that funding calls specifically encourage or require the use of human-specific non-animal technologies for human disease research and no longer accept or request funding applications that require the use of animal models. We would like to stress that funding should be made available to assess the capacity of the organoids in predicting patient responses or for evaluating novel, possibly combination therapies, and not for comparative studies aiming to ‘validate’ organoids against animals, given that the animals models are, and remain, unvalidated.

Interestingly, there is a strong reliance on charity donations for cancer research in the European Union. An analysis of sources of cancer research funding from 2006 revealed that around 50% originated from charities [21], whereas in the US, over 90% of cancer funding is public/governmental in origin [22]. In the European Union, charitable contributions made up EUR 667.3 million of cancer research funds in 2002/2003, compared to EUR 662.3 million from 74 governmental sources across 28 member states (not including European Commission funding). This reliance in Europe on individually donated funding adds pressure on the research to produce tangible outputs that will benefit the citizens donating and adds to the need for a measurable return on investment. The sources of funding have considerable significance for research—with different charities dedicated to specific cancer types or patient groups. A recent editorial posed the question whether “this is the most efficient means to support cancer research, given unavoidable duplication of administrative costs and loss of economy of scale” [23]. Of course, the pharmaceutical industry is also heavily invested in research and development for cancer; in 2008, the top 24 pharma companies contributed over USD 3 billion, accounting for almost one quarter of the total investment in global cancer spending [22]. Pharmaceutical companies are often criticized for what are perceived to be over-inflated drug prices, but to counter that, costs are high to support the R&D required to develop the transformative drugs needed. That said, an analysis of 26 such drugs showed that publicly-funded academic research was the primary source of drug innovation [24].

In an ideal world, drug costs would be low(er) and success rates would be high(er), but even in a less-than-ideal world, it seems that there are shifts in the drug development and testing paradigm that could improve the cost-efficiency of this process and therefore enable a healthier return on investment [25,26,27,28]. There are well-recognized issues with high attrition rates for all drugs [26,29,30], and for oncology, the likelihood of success is estimated at less than 4% [29]. This figure has remained largely unchanged for over 15 years [31]. The reasons behind this are many—initially it was thought that the rapid increase in the number of new oncology agents developed at the start of the 21st century contributed to increased drug failures [32]—but there are also market pressures, regulatory constraints, scientific challenges and, perhaps most vital, less than optimal preclinical testing strategies.

At the very least, it seems that human cell-based methodologies could be applied to enhance preclinical testing strategies for drug repurposing—particularly since this approach is showing promise for cancer [33]. Drug repurposing refers to the practice of applying marketed drugs for a different indication than the original approval and can accelerate clinical use, given the extensive safety and pharmacological data which are available. Using non-animal approaches, including phenotypic screening, computational approaches and advanced 3D human cell-based models, including PDO, for pre-clinical efficacy testing may enable promising non-oncology drugs, or novel drug combinations, to be fast-tracked into clinical trials for cancer [34].

With increasing public (and scientific) discomfort in research using animals and demands for ‘alternatives’ to animals to be used [35,36,37,38], it is timely to consider where the non-animal methods are and what is needed to implement these more effectively. Preclinical safety and efficacy testing continue to rely heavily on animal models and for cancer, it is mice who bear the brunt of such investigations (Table 4). As revealed by the 2019 report from the Commission to the European Parliament and the Council on the statistics on the use of animals for scientific purposes in the Member States of the European Union in 2015–2017, in the EU alone, around 1 million mice are used for cancer research every year, representing over 90% of the total animal use for cancer research. These data are the total number of mice recorded in the categories of oncology and human cancer and are likely to be an underestimate of total mouse use for cancer. They do not include numbers used for the preclinical testing of potential oncology drugs, as it is not possible to estimate those data from records submitted. Moreover, we do not have access to the number of mice used for any research in the US, since data published by the USDA exclude mice, rats, fish and birds.

Whether funding is publicly donated or taxpayer-derived, in the face of public opposition to animal research, research funding calls and awards should be more circumspect, and concerned with delivering tangible, relevant and translatable outputs and return on investment to taxpayers and charitable donors.

It is true that cancer drugs are a leading contributor to healthcare costs [39], and efforts are underway to associate clinical benefit to costs, in order to optimize access to beneficial drugs and ensure that limited public health resources are used to best effect. The European Society of Medical Oncology—Magnitude of Clinical Benefit Scale [40] and the American Society of Clinical Oncology Value Framework Net Health Benefit [41] were developed to guide the evaluation of the value of cancer treatments, partly in response to concerns about the rising costs of cancer drugs. Analysis of 65 FDA-approved drugs for solid tumors showed that there were no significant differences in costs according to clinical benefit, such that a drug with a lower clinical benefit score had similar or even higher monthly costs compared to higher-scoring drug [39]. Annual costs for cancer treatments may reach tens of thousands of US dollars per patient per year [42], and drugs used for multiple indications are costed higher when used for cancer, e.g., the same monoclonal antibody can be priced at over USD 100,000 a year more for hematology or oncology than for other diseases [43]. These increasing drug costs are adding to the overall public health burden, magnifying the stresses imposed on limited resources by the ageing population, increasing incidence in chronic diseases, and the impact of more years lived with disability or injury. Thus, it is imperative that research is optimized, to generate safe and effective drugs and to ensure that the costs of these therapies are adjusted relevant to their value—including health benefits, side effects, overall burden for patients receiving treatment, as well as the financial cost. To optimize research impact and expenditure, there is also a need to consider the impact of disease prevention—on reducing health costs, improving quality of life—and reducing animal use in research. During our analysis of clinical trials, we noted that around 13% of the interventional clinical trials for breast cancer were classed as ‘behavioral’ and focused on non-pharmacological approaches such as exercise, diet, mindfulness. Applying non-animal, human-relevant methods would help to better our understanding of the molecular, genetic, epigenetic bases of human disease [28] and therefore could identify effective therapies. Using epidemiological studies to monitor the impact(s) of diet, exercise, and other human lifestyle habits offers the opportunity to evaluate the impact of non-pharmacological interventions to potentially alter the course of disease progression, to reduce disease incidence, potentially reduce drug use and public health costs, and to improve the health of the population.

We noticed that the number of publications focused on the use of PDO is gradually increasing over time worldwide, for all three cancers studied here (Figure 3C). We also noted that, despite a lag in comparison to the animal-based methods, NIH support for new research projects that use PDO is gradually on the upturn (Figure 4E and Table 3), which is a promising indicator of a shift toward more cost-effective and human-relevant approaches to understanding human disease. PDO are valuable tools for cancer research and pre-clinical testing and their use could be further enhanced to facilitate drug screening [44]. Combining PDO systems with, for example, induced pluripotent stem cells, which can overcome some of the issues associated with primary cell culture, or with other human 3D culture platforms that incorporate dynamic culture conditions could offer more flexibility for evaluating tumor development, drug delivery and potential molecular targets [45].

In our analysis, the exception we see is a small drop between 2018 and 2019 for colorectal cancer, which is still the one type of cancer that has more papers that declared having received funds from NIH, if compared to breast and lung cancers. However, delivering more returns from the human cell-based methodologies will require more dedicated support, e.g., funding calls that encourage or require that researchers use non-animal methodologies which offer more human-relevance—such as organoids. It is disappointing that for the cancers we studied, only one study using PDO was associated with a clinical trial. This could be linked to the particular cancers that we studied, or it may indicate a reluctance to apply these relatively new, so-called animal replacement technologies to clinical study. We appreciate that for safety and efficacy testing purposes, there are legal requirements that mandate animal use, but this is not the case for basic or applied and translational research—the very reasons behind 70% of the mice used for scientific purposes in the European Union (Table 4). The Nuffield Council of Bioethics report on the ethics of research involving animals stated that “There are other possible constraints [beside scientific objections] that may impede the implementation of Replacements. They include: regulatory inertia, insufficient funding, non-availability of human tissue, lack of incentives to explore the potential of Replacements, lack in the availability of information about suitable Replacements, insufficient integration of in vitro and in vivo research, and the possibility that tradition and conservatism may mean that researchers are reluctant to explore the potential of Replacements” [46]. Several of these sentiments were echoed in the report of the European Commission’s ‘Non-animal approaches—the way forward’ scientific conference in 2016 [47]. There is a need to better understand the obstacles to using human-relevant approaches and to create strategic roadmaps to enhance their development and ensure their implementation, in order to ultimately improve human-relevant research [48].

In their editorial of 2011, Prinz et al. [49] commented on the plummeting success rates for Phase II trials, directly related to insufficient efficacy [50], and declared that “This indicates the limitations of the predictivity of disease models and also that the validity of the targets being investigated is frequently questionable…” The authors carried out their own analysis, revealing issues with reproducibility and failure to publish ‘negative’ data that can skew the research landscape and lead to needless duplicative studies, and ultimately called for more in-depth biological understanding of the target. To ensure that we are not misunderstanding human tumor biology, it seems important to take advantage of what the organoids, together with other human cell-based approaches, can offer. For example, genetic screening of colon cancer organoids indicated that they retain individual patient genetic diversity in culture, and thus could be used to enable the identification of gene-drug interactions, which could be used in medium- to high-throughput drug sensitivity screening (not currently possible with PDX models) to inform specific treatment strategies [51]. In their review of organoid use, Granat et al. also recognized the capturing of patient heterogeneity as an advantage of organoids not present in PDX models, and suggested that the shorter time required (weeks to establish PDO compared to months for PDX), lower costs and enhanced ‘longevity’ of PDO are attractive features that could promote the use of PDO in preclinical testing strategies [52]. Another advantage of organoids compared to the PDX models is the possibility to use organoids to create biobanks of PDO [53]. Importantly, the study by Sachs et al. compared drug responses in organoids to those of the patient and PDX models—showing that the in vitro method matched both whole animal and patient responses—evidence that the organoids could be used as a replacement strategy for the animal models [53]. The development of living biobank resources, where human tissue-based models can be shared ethically, openly and fairly, could address issues with tissue availability that prevent some researchers from using human systems [54,55], and represents a hugely important step toward the application of in vitro models as preclinical models to replace poorly performing animal surrogates.

## 5. Conclusions

In summary, we observed modest increases in outputs (research papers, research funding, clinical trials) related to the application of PDO for the three main types of cancer afflicting Western societies. However, we feel that more could be done to incentivize the use of such human-relevant methods, and call for the NIH, European Institutions and Member States, and other major funding organizations, to prioritize the use of human biology-based approaches such as organoids. This could involve not only targeted funding programs and calls designed to accelerate the development and use of human cell- and biology-based methodologies and related infrastructures in health research, but also the addition of directional language to all health research funding calls emphasizing human biology as the gold standard. There is a great need to level the ‘playing field’ for the human cell-based technologies compared to animals, given that, in the European Union, only 0.036% of the research budget is dedicated to these more human relevant, non-animal approaches [28]—this deficit is adding hugely to the obstacles preventing animal replacement in human health research. Additionally, we recommend establishing training programs for researchers to increase visibility and confidence in the applications of human-relevant methodologies, and more transparency regarding where organoids/other human cell-based tools are used to develop drug testing regimes for patients.

## Figures and Tables

**Figure 1 animals-10-01923-f001:**
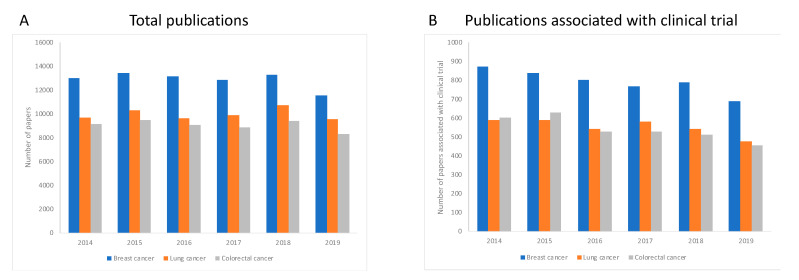
Publications for each type of cancer investigated, and associations with clinical trials. Panel **A** shows the total number of published studies retrieved from PubMed during the period 2014–2019, for each type of cancer, regardless of the technology used. For each of the three cancer types, the number of publications is reduced over time. Panel **B** shows that the same was true for studies associated with clinical trials, where the number of published studies associated with a clinical trial decreased for each cancer type between 2014 to 2019.

**Figure 2 animals-10-01923-f002:**
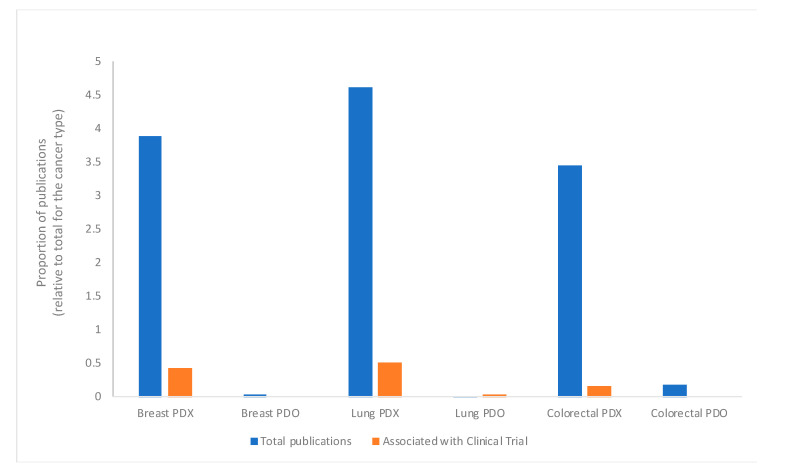
Publications employing patient-derived xenograft (PDX) or patient-derived organoids (PDO) for each cancer type, and their association with clinical trials. The proportion of papers published between 2014–2019 for breast, lung and colorectal cancers using PDX or PDO, relative to the total number of papers published for each cancer type, regardless of the technology used, and the papers associated with a clinical trial are shown. Blue bars indicate the proportion of papers using each technology, relative to the total number of papers for that specific cancer type, and orange bars indicate the proportion of papers associated with a clinical trial. Data were retrieved from PubMed as described in the Methods.

**Figure 3 animals-10-01923-f003:**
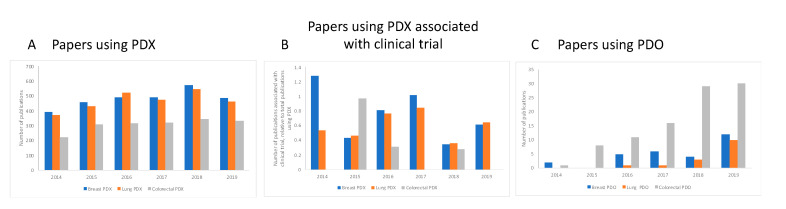
Year-on-year analysis of the number of papers published worldwide using PDX (**A**), the proportion of those papers that used PDX and were associated with clinical trials (**B**), and the number of papers published using PDO (**C**). The number of studies using PDX, for all three types of cancer, shows some fluctuations but overall remain the same over the time period evaluated (**A**). Likewise, there is no growing trend for the number of publications using PDX and associated with clinical trials between 2014 and 2019 (**B**)—these data are expressed as the number of studies associated with a clinical trial relative to the total number of publications using PDX. However, panel C shows that the number of published studies using PDO increase over time but note that the total number of papers is still low compared to the number of studies using PDX (panel A).

**Figure 4 animals-10-01923-f004:**
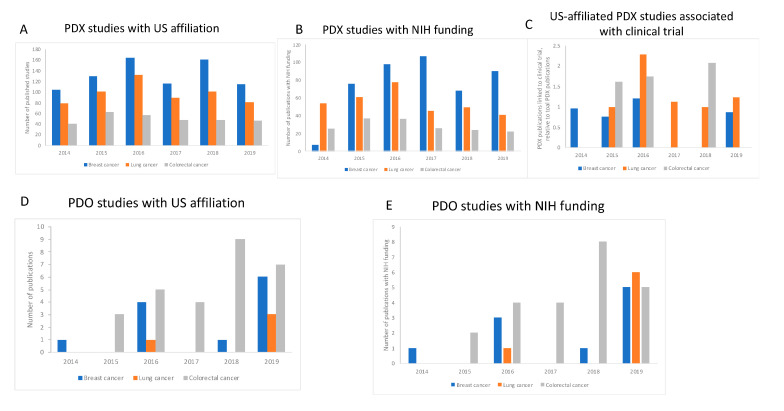
Research using PDX or PDO that declared an affiliation to the US. Panel **A**: The number of studies using PDX during the 2014-2019 period showed some fluctuations but overall remained the same during the period. Panel **B** shows the number of studies using PDX during the period 2014–2019 that received National Institutes of Health (NIH) funding. Note that the number of studies has not increased over the years. Panel **C**: The proportion of studies using PDX, relative to the total number of using PDX, which were associated with clinical trials, does not show a growing trend. Panel **D** indicates that research using PDO is on the increase in the US. Panel **E** shows the number of studies using PDO that received funds from NIH and indicates that most studies using PDO in the US are NIH-funded (or part NIH-funded).

**Figure 5 animals-10-01923-f005:**
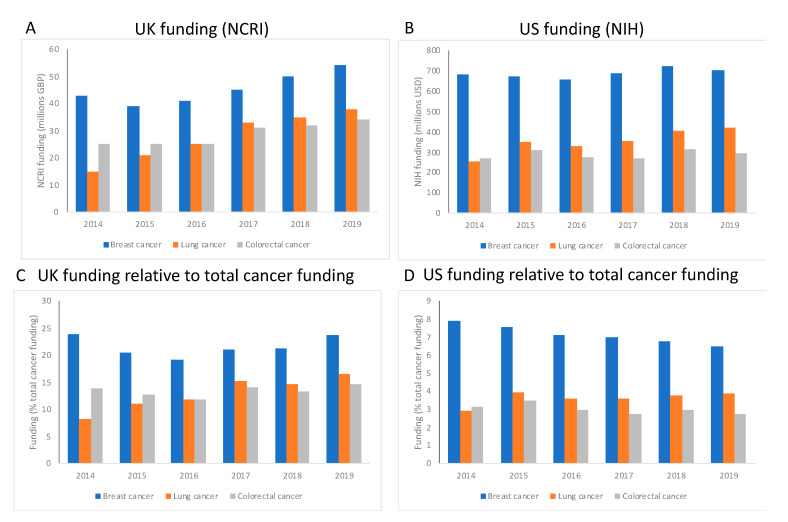
Funding for cancer research in the UK (panels **A** and **C**) and US (panels **B** and **D**) between 2014 and 2019. Data were retrieved from National Cancer Research Institute (NCRI) and NIH databases for the UK and US, respectively. Panels **A** and **B** show funding for each selected cancer type for the UK and US, respectively, revealing relatively generous support for breast cancer for both regions. Panels **C** and **D** are the funding for the selected cancer types relative to total cancer funding, for the UK and US, respectively. Support has been sustained or increased slightly over time for both regions.

**Table 1 animals-10-01923-t001:** Details of the databases used to gather information for this project. These databases are all publicly available and were accessed between February and September 2020 in completion of the research presented here.

Database	Details	Weblink
CORDIS	The Community Research and Development Information Service (CORDIS) is the European Commission’s primary source of results from the projects funded by the EU’s framework programmes for research and innovation (FP1 to Horizon 2020).	https://cordis.europa.eu/projects/en
Globocan	The Global Cancer Observatory (GCO) is an interactive web-based platform presenting global cancer statistics to inform cancer control and research.	https://gco.iarc.fr
The National Cancer Research Institute (NCRI)	NCRI has been collecting research funding data since 2002 in order to understand how money is distributed across various areas of research, and identify any gaps	https://www.ncri.org.uk/ncri-cancer-research-database-old/spend-by-research-category-and-disease-site/
National Institutes of Health (NIH) Data Book	The NIH Budget and Spending reports provide information about the detailed estimates and justifications for research and research supported activities along with a comprehensive overview of official presentations.	https://report.nih.gov/categorical_spending_project_listing.aspx?FY=2018&ARRA=N&DCat=Parkinson%27s%20Disease
NIH Research Portfolio Online Reporting Tools (RePORTER)	The Research Portfolio Online Reporting Tools provides access to reports, data, and analyses of NIH research activities, including information on NIH expenditures and the results of NIH supported research.	https://projectreporter.nih.gov/reporter.cfm
PubMed	PubMed is a free resource supporting the search and retrieval of biomedical and life sciences literature with the aim of improving health–both globally and personally. The PubMed database contains more than 30 million citations and abstracts of biomedical literature	https://pubmed.ncbi.nlm.nih.gov

**Table 2 animals-10-01923-t002:** Published studies which were funded by, or declared an element of funding by, the US National Institutes of Health (NIH) (excluding reviews) were retrieved from the PubMed database, using search terms defined in the methods, for models of three types of cancer. As an initial, broad measure of assessing impact, the PubMed filter for clinical trials (Article Type) was applied to determine the number of clinical trials associated with particular publications.

Topic of Study	Total Number of Publications 2014–2019	Number of Studies Associated with Clinical Trials	Number of Publications Stating NIH Funding	Number of Clinical Trials Associated with NIH-Funded Publications
Breast cancer—PDX	2622	18	466	3
Breast cancer—PDO	28	0	10	0
Lung cancer—PDX	2481	15	291	6
Lung cancer—PDO	14	1	10	1
Colorectal cancer—PDX	1648	5	159	3
Colorectal cancer—PDO	88	0	20	0

**Table 3 animals-10-01923-t003:** NIH research project funding, categorized as R01 and year 1 of support, retrieved from NIH RePORT (Table 1) using either breast cancer AND mouse AND xenograft or breast cancer AND organoid between 2014 and 2019. Grants were further scrutinized to ensure that breast cancer was the focus of the research, and that either xenografts or organoids were one of the methodologies described. Total funding for grants that fulfilled these criteria was calculated.

	2014	2015	2016	2017	2018	2019
Number of R01 grants returned with breast cancer, mouse, xenograft	17	20	19	26	22	23
Number of R01 grants returned with breast cancer, organoid	1	0	1	2	2	4
Total funding for R01 projects returned with breast cancer, mouse, xenograft (USD)	6,882,560	8,292,511	8,230,285	10,916,262	8,517,590	10,621,030
Total funding for R01 projects returned with breast cancer, organoid (USD)	315,367	N/A	392,078	750,642	1,099,033	1,791,059

**Table 4 animals-10-01923-t004:** The total number of animals used for scientific purposes throughout the European Union in 2015, 2016 and 2017, numbers of animals used in basic and translational/applied research and the numbers of animals used specifically in cancer research. Mice are the most commonly used species, and make up the majority of animals used for basic and translational/applied research and for cancer research. Data from [17].

Year of data Collection	2015	2016	2017
Total number of animals used for all scientific purposes	9,782,570	10,028,498	9,581,741
Total number of mice used for all scientific purposes	5,766,804	6,043,947	5,756,121
Relative burden on mice (number of mice used relative to total animal use for all scientific purposes)	59%	60%	60%
Total number of animals used in basic, translational and applied research *	6,665,081	7,042,567	6,557,609
Total number of mice used for basic and translational research	4,449,650	4,780,898	4,565,334
Relative burden on mice (number of mice used relative to total animal use for basic, translational and applied research)	67%	68%	70%
Total number of animals used in cancer research **	1,008,724	1,110,378	1,196,945
Total number of mice used for cancer research	961,971	1,059,036	1,103,749
Relative burden on mice (number of mice used relative to total animal use for cancer research)	95%	95%	92%

*—Does not include the animals used for regulatory purposes such as the toxicity and safety testing required for compound approval. **—Calculated by totaling number of animals/mice used for oncology research and number of animals/mice recorded as used for human cancer research.

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
