# Peer review of "Patient-Derived Xenograft vs. Organoids: A Preliminary Analysis of Cancer Research Output, Funding and Human Health Impact in 2014–2019"

_animals, 2020, doi:10.3390/ani10101923_

Round 1

Reviewer 1 Report

The authors have greatly improved their manuscript, which now is a very pleasant read. As I had many comments, I understand that they missed one, which still needs to be addressed:

New version lines 642-648: This is a new (albeit small) analysis with new results. It should be described in the methods and results sections, or deleted (preferable, read on why). Please note that your strategy here only retrieves trials that mention PDOs and PDXs in their trial registration. Most clinical trials will not mention any preclinical strategies in their trial registration, you probably only retrieved trials using PDOs and PDXs for individualized medicine trials. This results in an underestimate, which needs to be discussed if you keep this analysis in.

While I did not read too carefully for spelling and grammar (the paper is well-written and I think that copy-editing will be sufficient), but spotted a typo on line 193-194.

I look forward to seeing this paper in print and being able to share it with my colleagues.

Author Response

We thank the reviewer for their time and patience and for very constructive ,helpful comment that we feel have strengthened our paper.

We have gone through the paper again and used the inbuilt grammar and spelling check, and have been assured that the journal editors will also do this, so we trust that this will address any typos.

We agree with the concern raised (lines 2562-2568 below) regarding our attempts to draw associations between clinical trials and the different technologies - this is complex and probably warrants further, more specific analysis.  We have deleted that section as suggested.

We do hope that the reviewer will share this paper and thank them again for their valuable insight.

Reviewer 2 Report

Having carefully read this re-submission I feel that the authors have gone to considerable lengths to address the reviewers comments and incorporate suitable sections and alterations which improve the manuscript. Both minor grammatical and typographic errors have been fully addressed.

I do have, however, some concern about the points I raised under 'Additional Points Raised': 

5. They state in their response to reviewer 1 that they have added another reference (13)....Does this reference cover the topic of my original concern? 

Ref 13; Bleijs et al (2019) does not seem to cover the topic of differential funding through the charity sector!

6. While they have covered my point related to re-purposed drugs and speedy translation to clinic via NAMs, References 23 and 24 - as stated in their response to Reviewer 1 - do not relate to this area, rather, Refs 35 & 36 (as now used in the corrected manuscript) are appropriate.

Can the authors please confirm their error on the latter in their response to Reviewer 1 and check ref 13 against point 5 again please?

Other than this small matter, I would recommend publication now.

Author Response

We thank the reviewer again for their time and their helpful, constructive comments on the paper. We apologise profusely for the errors that appeared in our response to review document- the reviewer is completely correct that the references described therein are not appropriate and were an error on our part. Please be assured that these are not transferred to the manuscript - here we do have reference 23 and 24 for charity funding and references 35 and 36 for drug repurposing.

Thank you so much for pointing these out, we appreciate the efforts and the improvements that you have helped make to our paper.

This manuscript is a resubmission of an earlier submission. The following is a list of the peer review reports and author responses from that submission.

Round 1

Reviewer 1 Report

The paper submitted explores a timely issue related to the substantial improvements, development and sophistication of novel complex 3D in 'human relevant' in vitro models which may be used in biological research and pre-clinical therapeutic testing. In particular the paper, which focuses on the 3 most numerically prevalent cancers (Breast, Lung and colo-rectal) addresses an important and hithero largely unexplored area of:

a) grants awarded for patient derived xenografts (PDXs) animal models in comparison to those awarded for patient derived organoids (PDOs)

b) publications on PDXs compared to those on PDOs

c) Clinical application (eg trials) associated with or as a result of PDX and PDO acquired data respectively and thus impact on human health

d) Trends in the period 2014-2019 inc with respect to the above 3 criteria

While some substantive numerical conclusions concerning the landscape of animal replacement by so-called human relevant New Approach Methodologies (NAMs) re not possible in this study, the work is novel, informative and provides a sounder basis for the suggestion of adoption of such techniques in order to reduce the use of animals (particularly mice in this investigation) in cancer research and therapeutic testing as well as providing more accurate representation of the organ affected and cancer type along with high throughput, lower cost assaying.

The paper is generally well-written and suitably organised. 

There are, however, a number of minor corrections suggested as well as additional points raised and questions asked as detailed below:

Minor Corrections:

Page 1, line 17: "....human models cell based...." should read "...cell based human models.."

Page 1, line 36: "...it's timely to consider..." should be "...it is timely to consider..."

Page 2, line 58: "...(although spontaneous metastases occur very..." should read (although spontaneous PDX metastases occur very...).."

Page 2, line 76: "And there also.." should read "There also ......"

Page 2, line 87: "....multicellular architecture...." should read "....multicellular nature and histo-architecture...."

Page 4, line 181: "When then queried..."  should read  "We then queried..."

Page 9, Figure 5A X-axis should be NCRI not NIHR

Page 13, line 465: "...but counter that costs...." should read "...but to counter that, costs..."

Page 15, line 538: "II projects..." Should this be "II trials..."??

Page 16, line 566: "...NAMs...." NAMs..." should be introduced here (for the first time in text) as New Approach Methodologies

Additional Points Raised:

  1. Although this paper, understandably, concentrate soley on mice, it should be noted that PDX is also carried out on rats in some investigations.
  2. While PDOs are, indeed, valuable tools in the context of cancer research and pre-clinical testing the combined use of these systems with induced pluripotent stem cells (iPSCs) allows greater flexibility for testing molecular targets for therapy and different developmental stages of tissues and neoplasms. The organoid models may also need to be used in concert with other 3D human cell-based in vitro models in order to assay for both function and drug delivery (these system may therefore need to be dynamic rather than static in nature).
  3. While studying the most prevalent cancers (breast, lung and colo-rectal) is common sense, the real challenges may come with the tumours evolving from more complex organs such as the brain and, indeed, such organoid systems are beginning to come into use for glioblastoma were local invasion is the major therapeutic obstacle as well as in secondary cancers (from breast and lung for example) where perturbing seeding to and development within the complex brain is increasing as a possible therapeutic strategy.
  4. The latter part of the final paragraph of the introduction may be better placed in the discussion/conclusion.
  5. The sources of funding are of considerable significance particularly within the UK where there is a well developed charity sector including tumour type-specific charities which raise monies for specific forms of cancer as well as those which afflict different groups of people (age, gender etc) which may have greater or lesser sums of money to award to researchers. 
  6. It is, perhaps, more likely that repurposed and reformulated drugs will be approved for pre-clinical testing in NAMs/organoids and fast tracked into clinical trials and clinic than the novel and targeted agents and immunotherapy protocols. Moreover, these may be considerably cheaper and thus attractive to both patients and policymakers. 

[I would suggest that the authors incorporate some additional sentences into the text to allude to the various points made above.]

Questions:

  1. Within the Materials and Methods why was no mesh term used for breast cancer (page 3)?
  2. I found the search strategies somewhat difficult to decipher (page 4). could there be more obvious sub-headings to distinguish PDXs and PDOs?
  3. There also seemed to be some duplication in the search strategies (eg page 5)

In conclusion, once these minor points have received attention I would strongly support the publication of this work as it provides a new take on how we can better bridge the gap between pre-clinical research and cancer therapy in a high throughput, cost-effective, ethical and disease/tissue/species relevant nature based upon some interesting data mined from funding, research, publication and impact perspectives.

Reviewer 2 Report

The topic addressed is very timely and relevant and worthwhile. Specific comments have been included in the attached file. For the literature searches, I have invited an experienced librarian to give comments. As a general point it can be stressed more that there is much research waste that needs to be avoided. What can funders do specifically to make a change happen? E.g. stimulating and funding performing systematic reviews of the literature in order to critically analyse all evidence that is already out there, before embarking on new studies in in vitro models, in animals and in humans. What is the evidence for the chosen model system? And what about validation of replacement alternatives? How to do this reliably against human data instead of routinely used animal models. 

Reviewer 3 Report

The authors should be applauded for carrying out a large series of interesting desk studies, and transparently describing most of the limitations accompanying the methods. These important results should certainly be in the public domain, and are relevant to the audience of Animals. However, the manuscript needs substantial work to meet the standards expected for publication in this journal. I have listed my concerns and suggestions below, and look forward to seeing a new version. Keep up the good work and good luck!

Major concerns:

The paper lacks structure and is currently a tough read (even though the English is good). I kindly ask the authors to 1.) put the methods, results and discussion in the respective sections (I added a list below), 2.) restructure the methods and the results, using corresponding subheader numbering (in line with the Animals template) for these sections, making a clear distinction between literature and funding and additional smaller analyses, 3.) complete the methods section; which software did you use to do the calculations, how did you retrieve and store the data, create the plots, etc. and 4.) structure the discussion (e.g. in the format: short summary of results, comparison with literature, strong points, limitations, implications, but mainly making a coherent story out of it while leaving out the methods and results).

List of information in the wrong sections:

  • Line 103-118: results, methods and discussion.
  • Line 142-144: discussion
  • Line 192-195: discussion
  • Line 208-212: discussion
  • Line 257-259: methods
  • Lines 344-350: discussion
  • Lines 371-375: mainly methods
  • Lines 480-495: new results and methods. Please add short paragraphs on animal number estimates to methods and results.
  • Lines 530-533: yet another new small analysis and results. Please note that your strategy (which should be described in the methods) only retrieves trials that mention PDOs and PDXs in their trial registration. Most clinical trials will not mention the preclinical strategies there, you probably only retrieved trials using PDOs and PDXs for individualized medicine trials. This results in an underestimate (limitation to be added to the discussion)

The tables and figures all need to be revised.

  • For the figures, you are mainly reporting absolute numbers, which should generally not be in line charts (they imply mathematically derived numbers such as means and medians). My preference would be 2D stacked surface charts, but stacked bar charts are an alternative. Add titles to the individual panels to aid the readers. You did a lot of work, for a naïve reader the titles will help to understand what is where.
  • For figure 2: please facet-wrap the figure into two panels for overall vs. clinical trial, and adapt the axis for the trial one. Also, why Org instead of PDO?
  • For the figure 6 caption: the second “panel B” should probably be “panel D”.
  • Consider deleting figure 4 and most of the associated text (lines 311-317), it is very similar to figure 3 and a brief mention of this in the text would be sufficient.
  • For the tables: use your abbreviations in the headers, use layered headers so they remain legible, and remove information from the headers that applies to the entire table. For example for table 1, you could have a first column type of cancer, a second column PDX/PDO, a first row type of cancer, type of model, number of publications and a second row <empty>, <empty>, total, clinical trial related, NIH-funding, trial-related + NIH-funded. Also, replace studies by publications.
  • Transpose the tables listing years so we have the subsequent years from left to right as in the graphs.
  • The different search strings should be tabulated. It would also be good to tabulate the different databases with a description (at the very least add what the abbreviations stand for).

Minor comments:

  • Line 42: delete “understandably” (it disrupts the flow of the sentence without enough informative value to warrant this)
  • Line 46-47: explain what the Globocan database is or delete “according to the globocal database” and add a reference
  • Line 55-60 rephrase; split in multiple sentences and correct the singular-plural error
  • Line 100: outside the US, the abbreviation NIH may benefit from explanation
  • Line 172: something seems to be missing )(
  • Line 130 and 132 (and possibly elsewhere, please check thorughout): you mean publications, not studies
  • Line 270 ad elsewhere: you did a preliminary analysis, without statistics. I am happy that you did not use statistics here, but please avoid the word “significant”.
  • Line 273-274: so slight that I would not mention it
  • Line 278-281: I’d delete this section. If you find it particularly interesting, explain why.
  • Line 284-285: there are many PXD models, please rephrase.
  • Line 284-289: I’d consider deleting this section. Three types of cancer is too few to reliably say anything on an association or ranking.
  • Line 294-295: clinical trial-related publications (you do not know if publications were associated with particular trials, or did I miss something in the methods?).
  • Lines 356-357: rephrase (numbers of applications are low)
  • Line 429: “animal use” is too broad a term here, you only looked at mouse xenografts
  • Lines 444-446: organoid studies probably cost a lot less than xenograft studies. The discussion would benefit from an estimate of costs/ study and/ or numbers of studies performed from the budget.
  • Lines 468-469: I’m not sure what the authors mean here. In an ideal fantasy world, we could have all drugs for free and 100% success in development, but if they mean a realistic situation, I think a reference and rephrasing are needed.
  • Lines 526-529: I think it is a bit too early to conclude this, organoids are a relatively new approach and legislation probably plays a larger role than reluctance.
  • As a limitation, describe that you will have missed primary studies that were published combined with a literature review (because of your NOT review string).

And just as an educative note to the authors, for future literature searches I’d recommend not to use [All fields] searches. They retrieve studies where the search terms are part of e.g. the affiliation or author names, without added benefit. In PubMed, [tiab] retrieves all title, abstract and author-provided keywords, and [MeSH] all indexed terms, i.e. everything relevant that you can search (the full texts are not searched anyhow).
